# Tailoring atomic diffusion for in situ fabrication of different heterostructures

Hui Zhang[1,3], Tao Xu [1,3 ✉], Kaihao Yu[1], Wen Wang[1], Longbing He [1] & Litao Sun [1,2 ✉]

Atomic diffusion has been recognized as a particularly powerful tool in the synthesis of heterostructures. However, controlled atomic diffusion is very difficult to achieve in the fabrication of individual nanostructures. Here, an electrically driven in situ solid-solid diffusion reaction inside a TEM is reported for the controlled fabrication of two different hetero-nanostructures in the Ag-Te system. Remarkably, the morphology and structure of the as-formed heterostructures are strongly dependent on the path of atomic diffusion. Our experiments revealed that the surface diffusion of Te atoms to Ag nanowires leads to a core-shell structure, while the bulk diffusion of Ag atoms give rise to a $Ag_2Te$-Te segmented heterostructure. Heat released by Joule heating caused the surface diffusion process to be replaced by bulk diffusion and thereby determined the structure of the final product. Our experimental results provide an insight into solid-state diffusion reactions under an electric field and also propose a new process for the fabrication of complex nanostructures.

[1] SEU-FEI Nano-Pico Center, Key Laboratory of MEMS of Ministry of Education, School of Electronic Science and Engineering, Southeast University, Nanjing, China. [2] Center for Advanced Materials and Manufacture, Joint Research Institute of Southeast University and Monash University, Suzhou, China. [3] These authors contributed equally: Hui Zhang, Tao Xu. ✉email: xt@seu.edu.cn; slt@seu.edu.cn

Hetero-nanostructures comprising multiple functional components integrated together can combine the advantages of their individual components and often achieve synergistic properties beyond the functionality of a single component[1–3]. In this context, one-dimensional (1D) hetero-structures have attracted significant attention because of the unique physical and chemical properties arising from hetero-interfaces, which enable their applications in various fields, such as energy storage[4–6], catalysis[7–9], biomedical sensing[10,11], optoelectronic devices, etc. 1D hetero-nanostructures can be categorized into core-shell structures and segmented binary structures in terms of the morphology[12,13], which are used in different applications. Generally, the morphology of the hetero-structures is heavily determined by the synthetic method. Over the past decades, a variety of methods have been developed to synthesize 1D heterostructures, which fall into three main categories: heterogeneous nucleation and growth[14], diffusion and deposition[15], and melt-casting and electrospinning[16]. Among these methods, diffusion methods are particularly attractive in view of their relative simplicity. The formation of new materials or new phases by diffusion methods including simple heterogeneous atomic diffusion, Kirkendall effect, ion exchange, or phase transformation in solid-state can be used to prepare 1D heterostructures with improved interface and morphology control. Great efforts have been made to achieve controlled diffusion and heterogeneous structure conversion. For example, Chen et al.[17] developed a solution-phase in situ diffusion growth method with heating treatment to fabricate Fe(OH)$_3$@α-MoO$_3$ core-shell nanorods. Resasco et al.[18] demonstrated that controlled and uniform doping in a heterogeneous material (Ta-doped TiO$_2$) can be achieved using a solid-state diffusion approach based on atomic layer deposition. Zhang et al.[19] reported an electrically driven in situ cation exchange reaction for the fabrication of CdS-Cu$_2$S core-shell structured nanowires. These results indicate that the diffusion method is very promising for the fabrication of heterostructures. However, faced with the need to minimize the size of electronic and photonic devices, researchers have made great efforts to develop efficient and controllable synthetic methods for individual nanomaterials for functional applications. To date, the tailored diffusion of atoms and ions and the mechanism at the nanoscale of solid-state diffusion reactions are bottlenecks in the efficient synthesis of 1D heterostructures for highly integrated nanodevices.

Herein, we conduct in situ tailored diffusion experiments inside a transmission electron microscope (TEM) system while applying a voltage bias. Using this method, it is possible to efficiently control atom diffusion to selectively fabricate either a core-shell nanostructure or a segmented binary hetero-nanostructure in the Ag-Te nanowire (NW) system. Migration under an electric field is the main driving force that promotes atomic diffusion[20–22], which in turn, controls the direction of diffusion of Ag or Te atoms. Local temperature caused by Joule heating may strongly affect the mode of diffusion (bulk diffusion or surface diffusion) of particles owing to the different diffusion activation energies and ultimately results in the formation of different structures, including core-shell nanostructures resulted from surface diffusion and binary hetero-nanostructures resulted from bulk diffusion. Hence, our method makes it possible to fabricate complex nanostructures with controlled specific structures.

## Results

**In situ experiment**. Ag and Te NWs (Supplementary Fig. 1) used in this work were synthesized by a previously reported solvothermal chemical approach[23,24]. Electrically driven fabrications

of 1D heterostructures were conducted inside a TEM (FEI Titan 80–300 operated at 300 kV) equipped with a TEM and a scanning tunneling microscopy (TEM-STM) holder. As illustrated in Fig. 1a, Te NWs were supported on a TEM grid, and the Ag NW mounted on an STM probe was controlled to contact an individual Te NW, followed by the formation of different 1D heterostructure by applying a bias voltage.

It is rather remarkable that the morphology and structure of the as-formed heterostructures are tailored by the direction of applied voltage. Under the application of positive voltage to the Ag NW against the Te NW for a few minutes, as shown in Fig. 1b, a significant number of Te atoms migrate into the Ag NW and eventually form an Ag$_2$Te-Ag core-shell structure. In contrast, if the bias voltage is reversed, an Ag$_2$Te-Te binary heterostructure is formed ultimately (Fig. 1c). Because of the formation of a semiconducting Ag$_2$Te layer, a Schottky-type contact is formed at the Ag$_2$Te-Ag interface and the current-voltage (I–V) curve transforms from line shape to an S-shaped curve (Fig. 1d, e).

**Formation of core-shell structures**. In order to understand the mechanisms behind the transformation, we first observed how an Ag NW reacts to form a core-shell structure. Figure 2a–f present a typical series of TEM images showing the shape evolution of an Ag NW under a positive voltage of 0.3 V (Supplementary Movie 1). This dynamic process can be divided into two stages. In the first stage, small hill-like protrusions are formed on the Ag NW surface (red arrows in Fig. 2b, c), whereas no obvious changes are seen on the Te wire. Next, the hillocks grow and fuse into larger hills, eventually forming a clear core-shell structure (Fig. 2d–f). After the reaction, high-resolution transmission electron microscope (HRTEM) images and corresponding fast

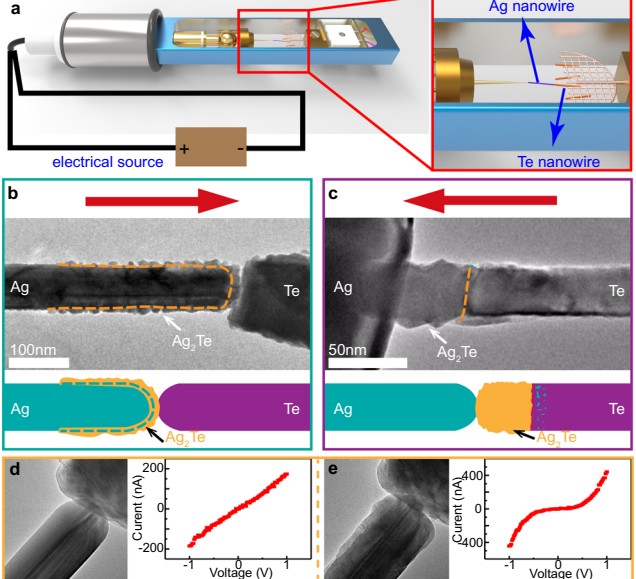

**Fig. 1 The formation of 1D heterostructures by applying a bias in TEM.** **a** Schematic diagram of the setup for in situ experiments. **b** The formation of the Ag$_2$Te-Ag core-shell structure by applying a positive voltage (red arrow showing the direction of current) to the Ag NW against the Te NW; the structure of the core-shell NW is illustrated at the bottom. **c** Low-magnification TEM image and the illustration showing the formation of Ag$_2$Te-Te binary heterostructure when applying a negative voltage. **d** I–V curve with the corresponding TEM image of the Ag-Te structure showing characteristics of an Ohmic contact before biasing; **e** I–V curve with the corresponding TEM image of the Ag$_2$Te-Ag structure showing semiconductor characteristic after biasing.

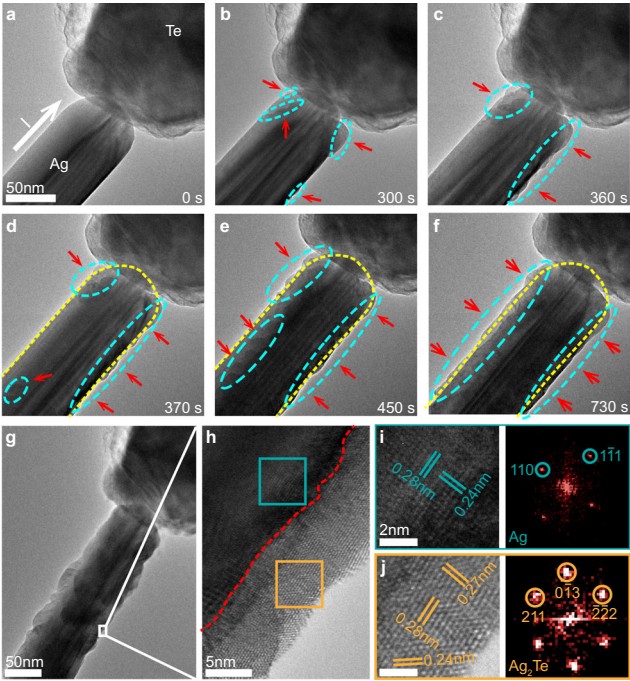

**Fig. 2 Microstructure and morphology evolution of an Ag NW under positive bias. a–f** TEM image series showing the formation of an Ag$_2$Te-Ag core-shell structure when 0.3 V bias was applied to the system. The red arrows and blue dashed circles indicate the growth of the Ag$_2$Te crystallite or shell, whereas the yellow dotted lines trace the shape of the Ag NW before biasing. **g** Low-magnification TEM image showing the final Ag$_2$Te-Ag heterostructure. **h** High-resolution TEM image of the area enclosed by the white square in **g** with the red dashed line highlighting the boundary between the Ag core and the Ag$_2$Te shell. **i, j** HRTEM images (left) and the corresponding FFT patterns (right) acquired from the core and the shell respectively.

Fourier transform (FFT) patterns were analyzed to determine the structure and composition of the products. Our results confirm that the core is made of Ag and the shell has the composition Ag$_2$Te (Fig. 2g–j) with the interface along with the red dashed line. The total reaction in this experiment can be expressed by the Eq. (1):

$$2Ag + Te \rightarrow Ag_2Te \qquad (1)$$

Negative Gibbs free energy ($\triangle G_f^0 = -45.4 \, kJ \, mol^{-1}$)[25] in this equation indicates that the reaction is spontaneous[26]. Owing to the low reaction barrier, Ag atoms could easily react with the diffused Te atoms, resulting in the formation of Ag$_2$Te crystals.

To investigate the process of formation of an Ag$_2$Te shell on the atomic scale, a series of HRTEM images were collected. Figure 3a–l shows the formation process of the Ag$_2$Te shell including the two stages described above (Supplementary Movie 2): nucleation and growth of small hill-like structures (Fig. 3a–f), and coalescence of hillock the hill-like structures and thickening (Fig. 3g–l), which is consistent with Fig. 2. Because of the electron-wind force and the low activation energy of surface diffusion, Te atoms are expected to diffuse into the Ag NW along the surface and congregate at areas where structural defects are present (Fig. 3b). Crystals of Ag$_2$Te nuclei form and grow into larger hill-like structures (Fig. 3c–f). The images suggest that the axial growth rate is much higher than the radial growth rate, which may be due to the fact that there is a sufficient number of Te atoms migrate from the Te NW on the surface and these react

easily with the Ag NW. Adjacent Ag$_2$Te hillocks grow and connect to each other, as shown in Fig. 3g–l, resulting in the formation of a stable shell. It is worth noting that there are more steps at the junction of two hillocks than in the middle of the individual crystallites. The greater number of steps and the faster growth can allow the hillocks to eliminate stress and connect into a stable shell structure (Fig. 3k), following which, the shell continues to thicken through a layer-by-layer growth (Fig. 3l). Moreover, the growth rate of the layer is closely related to the current and the nature of the contact. When a stable contact is obtained, the growth of the Ag$_2$Te shell is linear with time (growth rate is ~0.038 nm/s) under a constant bias (Supplementary Fig. 2), indicating that the preparation of these structures can be tailored by changing the applied bias under certain conditions. Our experimental results also reveal the relationship between the crystallographic orientations of the Ag$_2$Te-Ag core-shell NW (Fig. 3m), which corresponds to the structure in Fig. 3l. We identify that the orientation relationship is Ag$_2$Te{$\bar{1}$23}-Ag{111} due to the matching extent of the lattices (Supplementary Fig. 3).

In order to determine the extent of surface diffusion of Te, we performed energy-dispersive X-ray spectroscopy (EDX) analysis (Fig. 3n and Supplementary Fig. 4) in STEM mode. Figure 3n shows the EDX mapping of the front part of the core-shell structured NW; results confirm the distribution of Ag and Te elements. The intensity profile of Te along the radial direction is uniformly flat, whereas that of Ag shows a Gaussian distribution (Fig. 3o). This suggests that the Te element is only present in the uniform shell around the NW, which further demonstrates that the migration of Te into the Ag NW is predominantly through surface diffusion.

**Formation of segmented heterostructures**. It is rather remarkable that if a negative bias is applied, as shown in Fig. 4 (Supplementary Movies 3 and 4), it is not a core-shell but a segmented binary structure that is formed. In this case, Ag atoms migrate into the Te NW driven by the electron-wind force. In the initial stage, there is no obvious structural transformation, but a slight change, in contrast, is observed on the Te NW (Fig. 4a–c), indicating the diffusion of Ag atoms into the Te NW. After 80 s, structure clear phase transformation occurs (Fig. 4d and Supplementary Fig. 5), which can be identified by size changes as well as shape transformation, resulting from volume expansion accompanying the formation of Ag$_2$Te from Te. As time progresses, more and more Ag$_2$Te crystals are generated, and the change in size is much evident (Fig. 4d–f) with the diameter increasing by 15%, which is equivalent to the 30% increase in volume during structure change from Te to Ag$_2$Te. Results from the quantitative analysis on the formation process of the diffusion rate of reaction frontier and transformation rate of Ag$_2$Te as a function of reaction time are given in Supplementary Fig. 6. Similar to the generation process of the core-shell structure, the Ag$_2$Te phase transformation exhibits an approximately uniform velocity (~2 nm/s) with the constant current. However, the diffusion of Ag atoms becomes progressively slower with time (from ~8 nm/s to ~3 nm/s), which may be owing to the longer diffusion path. Notwithstanding, our results indicate that the rate of formation of the heterostructure can be tailored by changing the applied bias.

The chemical composition and structure of the reaction product were confirmed by HRTEM. As shown in Fig. 4g, there are three distinct areas in the as-formed heterostructure. The front part that is connected to the Ag NW has completely converted to Ag$_2$Te, whereas the part at the end still shows the crystal lattice of Te. It is noted here that there is a transition region between the two parts where the wire maintains the

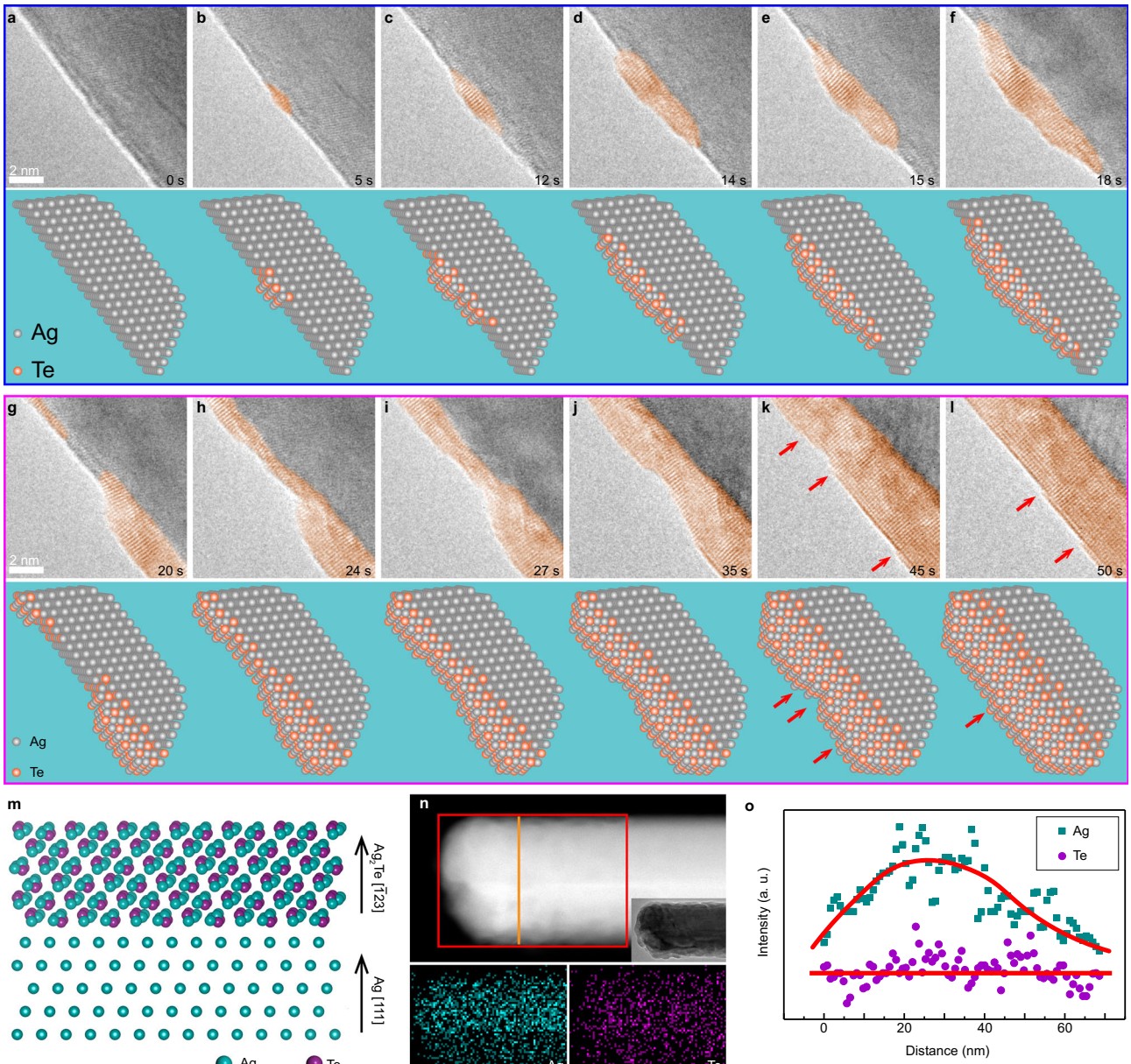

**Fig. 3 The growth process of Ag₂Te shell. a–l** TEM image series showing the growth of an Ag₂Te shell consisting of two distinct stages: nucleation and growth of small hillocks **a–f** and hillock coalescence and thickening **g–l**. The red arrows highlight the growth steps of the Ag₂Te layer. Below the TEM images are the corresponding structure illustrations. **m** Schematic diagram of the Ag₂Te-Ag heterostructure. **n** EDX elemental mapping images of Ag and Te corresponding to the red square area in the high angle annular dark-field (HAADF) image. **o** Intensity profiles of Ag and Te along the radial direction shown by the orange line in **n**.

original shape and the Te lattice framework although some new structures different from that of Te or Ag₂Te are seen. This indicates that the formation of Ag₂Te can be explained on the basis of atomistic bulk diffusion and chemical phase transformation. Under the influence of electric current, Ag atoms migrate into Te NW, after which the Ag₂Te phase transformation takes place when the diffused Ag atoms are sufficient. This explains why no obvious changes in shape and in size are seen in the initial stages of the reaction.

HAADF and EDX mapping (Fig. 4h) were obtained to analyze the chemical composition and elemental distribution in the Ag₂Te-Te heterostructure. It is noted that Ag-doped Te NW can easily break in areas near the phase transformation end owing to the low thermal evaporation temperature of Te (<300 °C)[27]. Our selected area is therefore the transition zone where a lot of Ag atoms have diffused into Te NW but have not yet completely converted to Ag₂Te, which means that this area is still in the diffusion stage (stage 1) of the solid–solid reaction. The intensity profile of both Ag and Te along the radial direction (Fig. 4i) presents a Gaussian distribution, indicating that the diffusion of Ag into Te is predominantly a bulk diffusion process. Moreover, a weak signal of the Te element was captured at a point in the front portion of the Ag NW, even though the atomic ratio of Te to Ag was determined to be only 1: (73.31 ± 35.27). (Supplementary Fig. 7). These results reveal that some thermal diffusion also occurs in the system. However, the electrically driven atomic diffusion evidently plays a major role in the transformation processes taking place in the Ag-Te system.

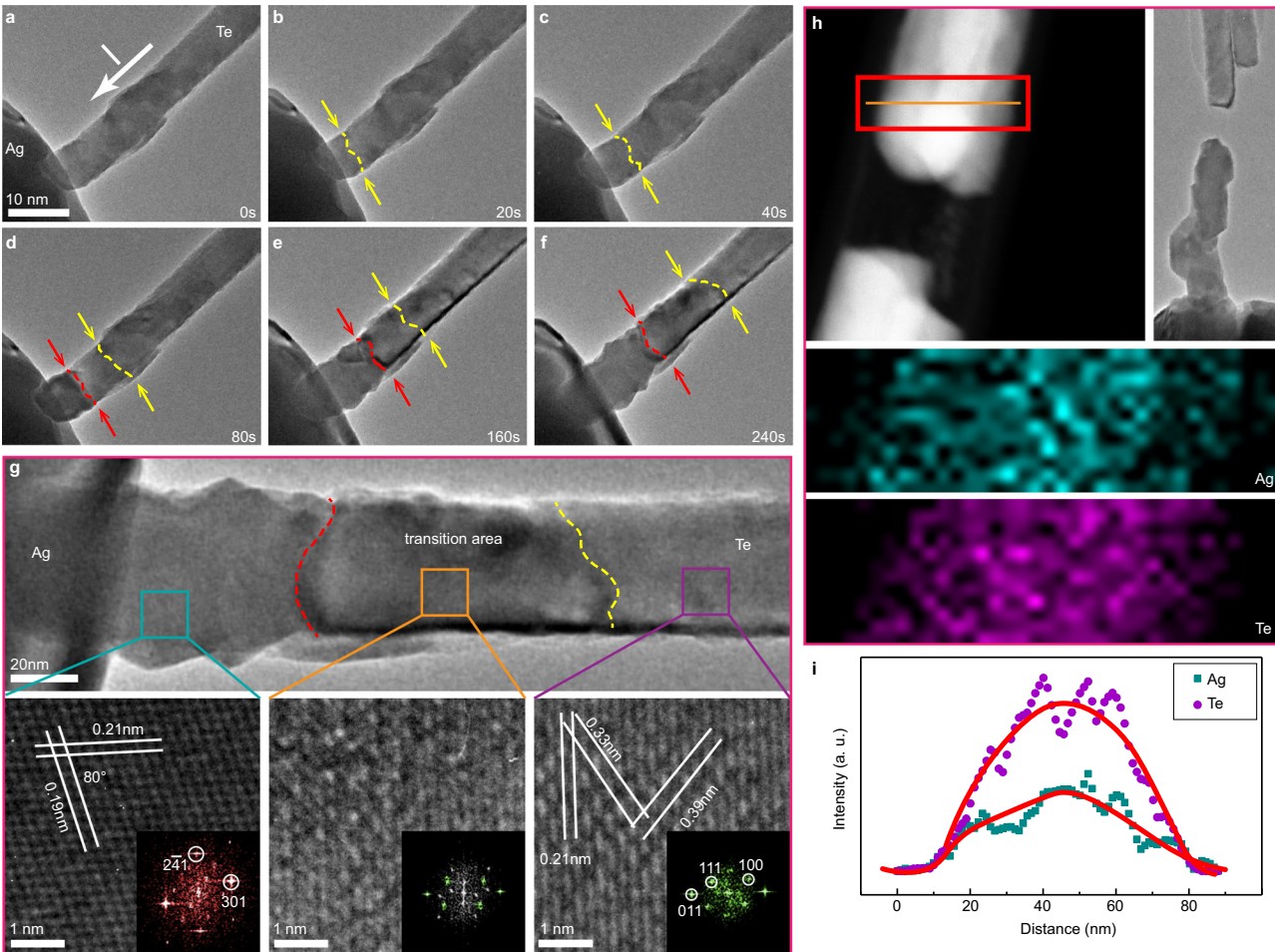

**Fig. 4 The formation of an Ag₂Te-Te binary heterostructure NW. a–f** TEM image series showing the growth of Ag₂Te under −0.8 V bias applied to an Ag NW. The yellow dashed lines highlight the diffusion frontier, and the red dashed lines indicate the phase transformation frontier. **g** TEM image showing the fully formed Ag₂Te-Te heterostructure in the Te NW. Shown at the bottom are HRTEM images and the corresponding FFT images acquired from different areas on the Ag₂Te-Te NW. **h** EDX elemental mapping images of Ag and Te corresponding to the red square area in the HAADF image. **i** Intensity profiles of Ag and Te showing the elemental distribution in the radial direction along the orange line in **h**.

## Discussion

Although electron beam irradiation may trigger atomic diffusion[28–30], the diffusion is not predominated by an electron beam in this work. Control experiments demonstrated that no obvious changes occur either in Ag or Te NWs even if the contacted system is exposed to electron beam without electrical bias for a long time (Supplementary Fig. 8). They are clearly different from the previous report that Ag₂Te NW could be formed only under electron beam irradiation when Ag and Te NWs were placed on the support film in close proximity[30]. The contrary phenomena should result from the different diffusion paths. When Ag and Te NWs are placed on a film, Ag species excited by electron beam irradiation could migrate in all directions on the film surface. In our experiments, Ag and Te NWs are suspended and point contacted, the excited species can only diffuse through the contact region, which will seriously limit the atomic diffusion and the formation of Ag₂Te. On the other hand, tens of experiments presented similar phenomena (Supplementary Figs. 9–11) that the atomic diffusion is strongly dependent on the direction of the electric field. However, electron beam-induced atomic diffusion should be isotropic, which indicates that the influence of electron beam can be ignored, and the diffusion should be driven by an electric field.

When a bias voltage is applied to Ag-Te NWs through the contact, collisions between the conduction electrons and the atoms can lead to the migration of the atoms. The driving force of electromigration is shown in Fig. 5a and Supplementary Fig. 12. For Ag and Te in this work, the atoms migrate in the direction opposite to the electric field[31,32]. When a negative voltage is applied to Te NW, Te atoms migrate into the Ag NW. Although there are other diffusion mechanisms (thermal diffusion or concentration gradient diffusion, Supplementary Fig. 7), we stress that electromigration has a significant role in this process because the diffusion direction is strongly dependent on the applied bias.

It is important to emphasize that the process observed in this work is not an electrochemical process. As reported in our previous work[19], electrochemical cation exchange reaction only occurs if the applied voltage is larger than the standard electrode potential of the material. In this work, atomic diffusion occurred even if the applied voltage (<0.5 V) is far lower than the corresponding standard electrode potential[33], $E^0(Ag^+/Ag) = 0.799$ V, $E^0(Te/Te^{2-}) = -1.143$ V, when both Ag and Te cannot be ionized electrochemically (Supplementary Fig. 13). It is also noted that the directions of atomic or ions diffusion are totally different. The direction of Cu⁺ diffusion driven by the electrostatic force in the electrochemical process is consistent with the direction of the electric field[19], whereas, in this work, the direction of atomic diffusion of Ag or Te atoms is completely opposite to the direction of electric field, as the driving force of the atomic diffusion is electron-wind force. Therefore, we consider that the process is

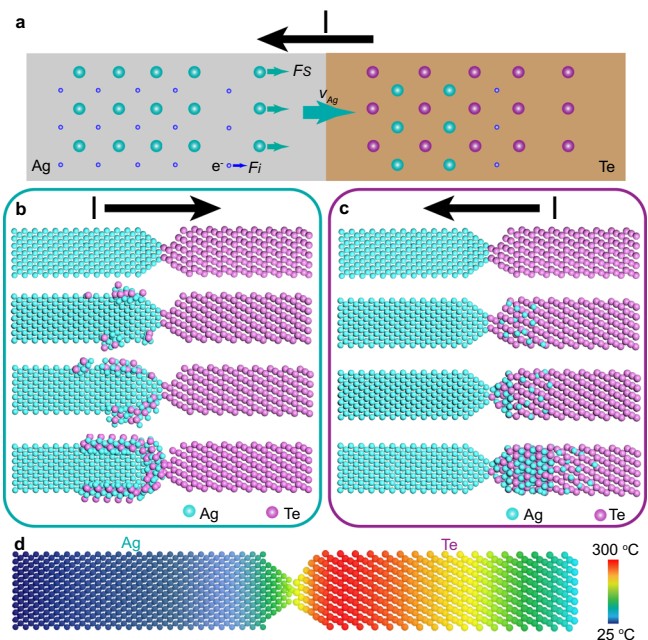

**Fig. 5 Formation mechanism of Ag-Te heterostructure. a** Schematic illustration of the forces acting on the Ag-Te system under a negative bias voltage. $F_i$ and $F_s$ separately indicate electrostatic forces and the ion or atom scattering forces (electron-wind force). **b, c** Schematic diagrams showing the formation processes of Ag₂Te-Ag core-shell structured NW and Ag₂Te-Te structured NW. **d** Schematic image showing the temperature distribution in the Ag-Te system.

electromigration-dominated diffusion followed by the spontaneous chemical reaction between Ag and Te[19,22].

As shown in Supplementary Figs. 10 and 11, no matter what conditions were, similar phenomena were observed and can be summarized in Fig. 5b, c. When a positive voltage is applied to the Ag-Te system, Te atoms diffuse into Ag NW along the surface under the influence of electron-wind force, and then the migrated Te atoms react with Ag to form Ag₂Te crystallite. Small pieces of Ag₂Te islands are connected into sheets, and finally, a shell layer is formed on the surface of the Ag NW (Fig. 5b). In contrast, Ag atoms begin to migrate into Te crystal by the path of bulk diffusion under a negative voltage bias. As shown in Fig. 5c, Te atoms and the migrated Ag atoms will undergo a process of phase transformation to form Ag₂Te when the atomic ratio is appropriate. Finally, the Te NW becomes an Ag₂Te-Te binary heterostructure. However, the current density might have an influence on the reaction speed. The reaction of thinner nanowires is faster than that of the thicker nanowires (Supplementary Fig. 10c, d, Fig. 2 and Supplementary Fig. 10b). It should be noticed that the formation of heterostructure is limited by the contact conditions, but it is hard to ensure the same contact in the separate experiment and the contact conditions are difficult to be accurately quantified inside the microscope, which makes challenges to reveal quantitative voltage-dependent kinetics from various Ag-Te systems (Supplementary Fig. 14).

The diffusion path may affect the final morphology of the as-formed heterostructures. Generally, surface diffusion takes priority over bulk diffusion in the solid materials owing to the lower diffusion barrier, however, bulk diffusion can be comparable if the system temperature is high enough when the thermal energy could help more atoms or ions to conquer the barrier of bulk diffusion.

We stress that temperature has a significant role in our process, and crucially affects the diffusion path, and eventually results in

the formation of heterostructures with entirely different morphologies[34–37]. We constructed a model to probe the temperature distribution of the Ag-Te 1D system under a bias (Supplementary Fig. 15 and Supplementary Note 1). When a voltage bias is applied to the 1D conductor system, heat will be generated along the conductor via Joule heating. Considering that the heat loss by radiative processes is negligible in vacuum (in a TEM), the heat can only be dissipated along the two contacted conductors through the contacts at their ends. The 1D conductor system can be set up to the state of thermal equilibrium within a short time[38].

For simplicity of calculation, we first consider the condition where the contact resistance is zero for the system. According to the general heat conduction equation, the maximum temperature rise by Joule heating can be estimated by[38]

$$\triangle T = T_{\max} - T_0 = \frac{\sigma U^2}{8\kappa} \tag{2}$$

where $U$ is the voltage drop on the NW, $T_0$ is the temperature of contact electrodes (room temperature). $\sigma$ and $\kappa$ are electrical conductivity and thermal conductivity respectively, which are assumed to be constant. It should be noticed that the contact resistance between Ag and Te nanowire is typically large owing to the amorphous carbon layer and could not be neglected. Depending on various conditions of contact, the contact resistance may range from $10^4$ to $10^9$ $\Omega$[39,40] and the heat generated from that will be large due to Joule heating. In consideration of the large difference of thermal conductivity between Ag (429 W m$^{-1}$ K$^{-1}$) and Te (2.35 W m$^{-1}$ K$^{-1}$), the side of Te NW will show a much higher and slowly changing temperature while the temperature reduces to the room temperature in a very short distance in Ag NW, as shown in Fig. 5d.

Applying typical parameters[27] of Te to Eq. (2), the maximum temperature rise of Te NW can be estimated to be over 300 K, which is close to the melting temperature of Te[41]. If the Umklapp phonon–phonon scattering process and contact resistance are considered, the maximum temperature $T_{\max}$ is expected to be higher[42]. Meanwhile, owing to the low thermal conductivity of Te, the heat generated by the contact resistance can promote the temperature of Te side when the heat is conducted along the Te NW. Once the Ag atoms diffuse into Te NW owing to the electromigration, the large thermal energy on the Te NW could be enough for Ag atoms to conquer the barrier of bulk diffusion in Te. Consequently, bulk diffusion is substantial for Ag atoms in Te NW, resulting in the formation of binary hetero-structured NW. In contrast, use typical parameters for Ag, we can estimate the maximum temperature rise at Ag NW to be <$10^{-2}$~10 K[43]. For the diffused Te atoms in Ag NW, surface diffusion still dominates over bulk diffusion due to the low barrier of surface diffusion, resulting in the nucleation and growth of a shell on the surface of Ag NW. Notably, the temperature distribution of the system is the same when the direction of current changes.

Following this analysis, this strategy can also be carried out in other materials. The direction of atomic diffusion can be tailored by current, whereas the heat generated by Joule heating can alter the path of atomic diffusion. Although other methods can produce well-defined heterostructures[44], selective customization for individual nanostructures still remains challenging. Using this method, controlled selective synthesis of individual heterostructure could be achieved by simply adjusting the current and local temperature, which is critical to tailor nanostructures and for the construction of future nanodevices.

In summary, an electrically driven solid–solid diffusion-reaction in the Ag-Te NW system was investigated using in situ TEM. The atomic diffusion process can be tailored by voltage bias in order to selectively synthesize individual hetero-

nanostructured NWs. The reaction mechanism is strongly supported by the fact that the diffused atoms are mainly driven by electron-wind force along the direction electrons flowed, whereas the phase transformation occurs spontaneously with an appropriate atomic ratio of Ag and Te. Moreover, the diffusion path can be tailored by local temperature caused by Joule heating, resulting in the formation of heterostructures with distinct morphologies. Surface diffusion of Te atoms in Ag NWs leads to the core-shell structure, whereas bulk diffusion of Ag atoms in Te NWs results in a segmented binary hetero-structured NW. Our results provide a new perspective to synthesis individual nanostructures with a diffusion-controlled reaction and also throw light on the micro-mechanism of solid–solid processes.

## Methods

**Preparation of Ag NW and Te NW**. Both Ag and Te NW samples were prepared by a solvothermal chemical approach. Ag NW preparation: 10 mL of ethylene glycol (EG, AR, 98%, Aladdin) and 1.5 g of polyvinyl pyrrolidone (PVP, molecule weight ≈ 24,000, Aladdin) was injected dropwise into 10 mL of a magnetically stirred solution of 0.1 M silver nitrate (AgNO$_3$, AR, 99.8%, Aladdin). The solution was then transferred to a Teflon-lined autoclave tube. The tube was sealed and heated at 160 °C for 2.5 h. The reaction mixture was cooled naturally to room temperature and the products were washed several times with acetone and water. Te NW preparation: 0.5 g tellurium dioxide (TeO$_2$, 99.99%, Aladdin), 1.5 g PVP (molecule weight ≈ 24,000, Aladdin), 7.5 mL NH$_3$·H$_2$O (ACS, 28.0–30.0%, Aladdin) and 1.625 mL hydrazine monohydrate (>98.0% (T), Aladdin) were dissolved in 110 mL deionized water under vigorous stirring and then transferred to a Teflon-lined stainless steel autoclave. The sealed autoclave was maintained at 200 °C for 16 h, followed by cooling naturally to room temperature. The products were washed five times alternately with ethanol (ACS, Aladdin) and water.

**In situ TEM experiment**. A half Cu grid, onto which a drop of a solution of as-prepared Te NWs was placed, was glued onto a gold wire using conductive epoxy. During this process, some Te NWs were anchored at the edge of the half Cu grid. W tips were prepared by electrochemically etching with NaOH solution (2 M) under a voltage of 2 V. All the W tips were cleaned by plasma cleaning for 60 s before use. The W tips were then soaked in the solution of as-prepared Ag NWs for 60 s at the end of which, several Ag NWs adhered to the surface of the tip. The setup of the in situ electrical experiments is shown in Fig. 1a. All the in situ experiments in this work were conducted in vacuum.

**Characterization of the structures**. The NWs and as-prepared hetero-nanostructures were characterized and analyzed using an aberration-corrected TEM (Titan 80–300, FEI). The electrical property measurements were carried out using a TEM-STM holder (PicoFemto). Dark-field images and EDX mapping were performed in STEM mode.

## Data availability

All the data supporting the findings of this study are available within this article and its Supplementary Information files. All raw images and source data are available from the authors upon reasonable request.

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

## Acknowledgements

This work was supported by the National Science Foundation of China (no. 61601116, 61974021, 51420105003), the national science fund for distinguished young scholars (no. 11525415), the Natural Science Foundation of Jiangsu Province (BK20181284), and the China Scholarship Council.

## Author contributions

H.Z. and T.X. conceived and designed the project. H.Z. and L.H. performed the synthesis of materials. H.Z. and T.X. performed the in situ TEM experiments. H.Z. and T.X. carried out the data analysis. H.Z., T.X., K.Y., W.W., and L.S. discussed the results. H.Z., T.X., and L. Sun wrote the manuscript.

## Competing interests

The authors declare no competing interests.
