## [Peer Review File · Nature Communications]

Reviewers' comments:

Reviewer #1 (Remarks to the Author):

This manuscript images the fusion of Ag and Te nanowires using in situ transmission electron microscopy. An applied electric field is used to induce the diffusion of either Ag into the Te nanowire or Te into the Ag nanowire leading to different kinds of Ag/Ag₂Te/Te heterostructures. Understanding the role of atomic diffusion in the creation of nanoscale heterostructures is critical for controlling the spatial arrangement of components. The in-situ imaging described in this manuscript provides important insights into these processes. However, the electron beam used for imaging may be a significant perturbation that also plays a role in the resulting heterostructures. The electron beam can reduce Ag⁺ ions as well as heat the sample. I recommend this manuscript for publication in Nature Communications after the Authors address the influence of the electron beam as well as other minor issues detailed below.

1) The Authors state on line 69 of p. 4 that, "Migration under an electric field is the main driving force that promotes atomic diffusion..."

However, electron-beam irradiation has also been shown to promote atomic diffusion in related heterostructures including Ag/Ag₂S and CdS/Cu₂S (see references below). The interaction of the electron beam with the sample should be taken into consideration in the analysis of how temperature controls the diffusion of Ag and Te atoms.

L. Motte & J. Urban, Journal of Physical Chemistry B, 2005, 109, 21499-21501.
H. Zheng et al., Ultramicroscopy, 2013, 134, 2017-213

2) The Authors should report the number of nanowires that were observed for each of the voltage conditions.

3) The signal-to-noise ratios of the EDX spectra shown in Supplementary Figures 4 & 7 appear to be relatively low. However, the compositions are given to high precision. For example, in Figure S4, the ratio of Ag to Te = 2.81 to 1, and compositional ratios with similar precision are given throughout the text. The Authors should justify this precision with the error associated with the EDX measurement.

4) There are some minor grammatical errors in the manuscript including the following:

The dashed circles in Figure 2 are blue whereas the caption refers to orange dashed circles.

Supplementary Figure 7 caption: change "atomic ration" to "atomic ratio" on line 82.

Lines 264 – 265 on p. 16 of the main manuscript: "Generally, surface diffusion is extremely prior in solid materials due to the lower diffusion barrier..."

as well as line 303 on p. 18, "Surface diffusion is still prior for Te atoms in Ag NW..."

I think the Authors mean surface diffusion dominates or takes priority over bulk diffusion in these statements, but the meaning of these statements should be clarified.

Reviewer #2 (Remarks to the Author):

In this paper, Sun and his coworkers put two metallic nanowires into contact and run the electrically driven solid-state chemical reaction to generate metal-semiconductor heterostructures. Although this is an interesting demonstration, it does not provide sufficient progress in the field to be published in Nature Communications. This is particularly true considering the same authors published a similar experiment with cation exchange reaction of cadmium sulfide nanoparticle in electrochemical TEM holder (ref. 17).

The paper claims the electrical bias-dependent formation of heterostructure (segmented structure vs. segregated core-shell structure) with surface diffusion of Te or bulk diffusion of Ag. The origin of the difference was speculated based upon the Joule heating phenomenon based upon the thermal conductivity difference. However, it would be more intuitive to discuss that bulk diffusion occurs because the more oxidizing Ag nanowire was on the anode side, and the more reducing Te nanowire was on the cathode side. Therefore, it would not be surprising that the segregated core-shell structure formed if we reverse the bias direction.

It would have also been great and scientifically novel if the paper discussed the more quantitative approach on the microscopic analysis of the nanostructure formation by controlling the voltage to derive another electrochemical kinetic model for the atomic control of nanostructures. Based upon the reading, I believe the paper would be more suitable for the more specialized journal in nanotechnology, such as ACS Applied Nanomaterials, etc.

Responses of the Authors to the Referee's Comments:

Reviewer #1:

This manuscript images the fusion of Ag and Te nanowires using in situ transmission electron microscopy. An applied electric field is used to induce the diffusion of either Ag into the Te nanowire or Te into the Ag nanowire leading to different kinds of Ag/Ag₂Te/Te heterostructures. Understanding the role of atomic diffusion in the creation of nanoscale heterostructures is critical for controlling the spatial arrangement of components. The in-situ imaging described in this manuscript provides important insights into these processes. However, the electron beam used for imaging may be a significant perturbation that also plays a role in the resulting heterostructures. The electron beam can reduce Ag⁺ ions as well as heat the sample. I recommend this manuscript for publication in Nature Communications after the Authors address the influence of the electron beam as well as other minor issues detailed below.

1) The Authors state on line 69 of p. 4 that, "Migration under an electric field is the main driving force that promotes atomic diffusion..."

However, electron-beam irradiation has also been shown to promote atomic diffusion in related heterostructures including Ag/Ag₂S and CdS/Cu₂S (see references below). The interaction of the electron beam with the sample should be taken into consideration in the analysis of how temperature controls the diffusion of Ag and Te atoms.

L. Motte & J. Urban, Journal of Physical Chemistry B, 2005, 109, 21499-21501.

H. Zheng et al., Ultramicroscopy, 2013, 134, 2017-213

Authors' Reply:

Thank you for the insightful comment. Without doubt, electron beam irradiation effects may affect in situ TEM experiments. We have conducted experiments to investigate the interaction of the electron beam with Ag-Te system. As shown in Supplementary Figure 8, there is no obvious changes either in Ag or in Te nanowires when the contacted Ag-Te system is only under electron beam irradiation for long time. Moreover, tens of experiments show that the atomic diffusion behavior is strongly depended on the direction of electric field while electron beam irradiation induced atomic diffusion should be isotropic. These results indicate that the influence

of electron beam irradiation effects can be ignored in our experiments. We have added some discussions in the revised manuscript.

Supplementary Figure 8. In-situ experiments only under electron beam irradiation without voltage bias.

2) *The Authors should report the number of nanowires that were observed for each of the voltage conditions.*

Authors' Reply:

Thank you for the instructive suggestion. In fact, we carried out tens of experiments to investigate the characteristics of the reactions. The statistics of the nanowires and the voltage conditions in the experiments has been presented in revised supplementary information (Supplementary Figure 9).

Supplementary Figure 9. Number of nanowires that were observed for each of the voltage conditions.

3) The signal-to-noise ratios of the EDX spectra shown in Supplementary Figures 4 & 7 appear to be relatively low. However, the compositions are given to high precision. For example, in Figure S4, the ratio of Ag to Te = 2.81 to 1, and compositional ratios with similar precision are given throughout the text. The Authors should justify this precision with the error associated with the EDX measurement.

Authors' Reply:

Thank you for the comments. The “Quantification Results” of the EDX spectra in Supplementary Figures 4 & 7 has been shown below. We have corrected the atomic ratio with the error in the main manuscript and Supplementary Information as well as below.

	Element	Weight %	Atomic %	Uncert. %	Detector Correction	k-Factor
Supplementary Figure 4	Ag(K)	70.36	73.74	9.34	0.89	7.089
	Te(K)	29.63	26.25	1.94	0.70	14.061
Figure 4h	Ag(K)	14.45	16.66	1.93	0.89	7.089
	Te(K)	85.54	83.32	3.26	0.70	14.061
Supplementary Figure 7b	Ag(K)	67.34	70.92	5.58	0.89	7.089
	Te(K)	32.65	29.07	2.34	0.70	14.061
Supplementary Figure 7c	Ag(K)	97.95	98.26	1.63	0.89	7.089
	Te(K)	2.04	1.73	0.81	0.70	14.061

Table 1. The quantification results of the EDX spectra in the Figure 4h, Supplementary Figure 4 and 7. The atomic ratio of Ag to Te in the Ag₂Te-Ag

core-shell structure is $(2.85 \pm 0.57):1$ in Supplementary Figure 4. The atomic ratio of transition area in the $\text{Ag}_2\text{Te}-\text{Te}$ structure is $(0.20 \pm 0.03):1$ (Figure 4h), while that of the completely reacted area is $(2.47 \pm 0.39):1$ (Supplementary Figure 7b). A weak single signal of Te element can be captured in the front point part of the Ag nanowire, where the atomic ratio of Te to Ag is only $1: (73.31 \pm 35.27)$.

4) *There are some minor grammatical errors in the manuscript including the following:*

The dashed circles in Figure 2 are blue whereas the caption refers to orange dashed circles.

Supplementary Figure 7 caption: change “atomic ration” to “atomic ratio” on line 82.

Lines 264 – 265 on p. 16 of the main manuscript: “Generally, surface diffusion is extremely prior in solid materials due to the lower diffusion barrier...”

as well as line 303 on p. 18, “Surface diffusion is still prior for Te atoms in Ag NW...”

I think the Authors mean surface diffusion dominates or takes priority over bulk diffusion in these statements, but the meaning of these statements should be clarified.

Authors' Reply:

Thanks for the valuable comments. We have checked the manuscript and corrected the grammatical errors in the revised manuscript.

Reviewer #2:

In this paper, Sun and his coworkers put two metallic nanowires into contact and run the electrically driven solid-state chemical reaction to generate metal-semiconductor heterostructures. Although this is an interesting demonstration, it does not provide sufficient progress in the field to be published in Nature Communications. This is particularly true considering the same authors published a similar experiment with cation exchange reaction of cadmium sulfide nanoparticle in electrochemical TEM holder (ref. 17).

The paper claims the electrical bias-dependent formation of heterostructure (segmented structure vs. segregated core-shell structure) with surface diffusion of Te or bulk diffusion of Ag. The origin of the difference was speculated based upon the Joule heating phenomenon based upon the thermal conductivity difference. However, it would be more intuitive to discuss that bulk diffusion occurs because the more oxidizing Ag nanowire was on the anode side, and the more reducing Te nanowire was on the cathode side. Therefore, it would not be surprising that the segregated core-shell structure formed if we reverse the bias direction.

It would have also been great and scientifically novel if the paper discussed the more quantitative approach on the microscopic analysis of the nanostructure formation by controlling the voltage to derive another electrochemical kinetic model for the atomic control of nanostructures. Based upon the reading, I believe the paper would be more suitable for the more specialized journal in nanotechnology, such as ACS Applied Nanomaterials, etc.

Authors' Reply:

We took into account all the comments and thank you for giving us a chance to "...provide unambiguous support for the electric bias-dependent morphology control and Joule heating-based mechanism. Alternatively, to provide additional novelty...". Here we can clearly explain the differences of the previous work (*Nature Communications*, 2017, 8, 14889) and current work as well as the related mechanisms.

Our current work is completely different from the previous work in mechanism: the previous work focused on the cation exchange reactions and is an electrochemical process in which Cu atoms were ionized electrochemically and exchanged Cd^{2+} ; but the present work is trying to use the effect of electron-wind force to control the atomic diffusion, where Ag and Te atoms (was not ionization electrochemically) diffused into another material and a subsequently spontaneous combination reaction under an appropriate atomic ratio. Supplementary Figure 11 shows the mechanisms of the two works.

The previous work (see the Supplementary Figure 11a below) focused on the research

of cation exchange reactions, in which one type of cation ligated within an intact anion sublattice (Cd^{2+} in CdS) are substituted by another kind of cation (Cu^+). The cation exchange reactions is an electrochemical process in which the Cu electrode is ionized electrically by applying a high voltage (which is higher than the standard electrode potential of Cu), after which the Cu^+ ions migrate towards CdS nanowire driven by the external electric field to react with the sulfide ions. The total reaction can be expressed by the equation: $2\text{Cu}^+ + \text{CdS} + 2\text{e}^- \rightarrow \text{Cd} + \text{Cu}_2\text{S}$.

But in this work, we developed a method to tailor atomic diffusion by using electron-wind force under a low voltage. The Ag_2Te nanocrystals was formed in this work even if the applied voltage (< 0.5 V) is far lower than the corresponding standard electrode potential, $E^0(\text{Ag}^+/\text{Ag}) = 0.799$ V, $E^0(\text{Te}/\text{Te}^{2-}) = -1.143$ V (*Lange's Handbook of Chemistry*), when both Ag and Te cannot be ionized electrochemically. The statistics of the nanowires and the voltage conditions in the experiments has been added as the Supplementary Figure 9 in the revised manuscript and supplementary information. The analysis indicates that the process in the present work is not electrochemical process.

It is important to emphasize that the driving forces of the atomic diffusion and cation exchange are completely different between the two works. As shown in Supplementary Figure 11, the direction of Cu^+ migration driven by the electric field force is consistent with the direction of the electric field in the previous work, while in this work the direction of atomic diffusion of Ag or Te atoms is opposite to the direction of electric field, as the driving force of the atomic diffusion is electron-wind force. Moreover, we also found that the changes in morphologies of the nanowires on the anode side are always more significant. But both anode and cathode materials kept their initial morphologies during the cation exchange reaction. These facts indicate that the reaction in this work is not electrochemical process.

On the other hand, electrochemical kinetic model cannot explain our present experiments totally. When Ag nanowire was on the cathode side and Te nanowire was on the anode side, it is difficult to explain the fact that Ag atoms still diffused into Te nanowire and further formed Ag_2Te crystals. In this situation, Ag nanowire on the cathode side would be protected by the external electric field and could not be ionized electrochemically. Meanwhile, the Te nanowire was on the side where materials tend to lose electrons. It means that the reaction between Ag and Te would hardly occur in this condition. However, we have seen that Ag atoms from the cathode side diffused to the anode side and further react to form Ag_2Te crystals, which is hardly explained by electrochemical kinetic model.

In conclusion, we consider that electromigration mechanism are more suitable to explain the atomic diffusion process in our work. The mechanism in current work consists of an atomic diffusion process driven by electron-wind force. The

experiments can be divided into two processes: the first electromigration process (physical) and the next Ag_2Te formation process (chemical). In the first electromigration process, the applied electric field plays a role to transfer the atoms and is a completely physical process. The temperature difference between two nanowires due to distinct thermal conductivities results in a different pathway of atomic diffusion. The diffused Ag atoms in the Te nanowire could gain enough energy from high temperature due to Joule heating, resulting in a high probability of atoms conquering the bulk diffusion barrier. In this situation, the occurrence of bulk diffusion can be compared to that of surface diffusion, therefore the diffusion of Ag atoms in Te nanowire presents the phenomenon of bulk diffusion in general. In contrast, the temperature of Ag nanowire still keeps low and the diffused Te atoms migrate in the mode of surface diffusion. Then, the diffused atoms react with another element spontaneously to form Ag_2Te crystals, when the atomic ratio of Ag and Te reached in an appropriate level (analysis and distribution of the temperature in a suspended 1D conductor system can be found in Figure 4, Supplementary Figure 12 and the last part in Supplementary Information). It is important to emphasize that the reaction barrier of Ag and Te is very low and the Gibbs free energy is negative ($\Delta G_f^0 = -45.4 \text{ kJ/mol}$) in this reaction, which indicates that Ag atoms could easily react with Te atoms, resulting in the formation of Ag_2Te crystals (*Journal of Phase Equilibria*, 1991. 12, 56-63; *Chemistry of Materials*, 2015. 27(15), 5189-5197; *Journal of the American Chemical Society*, 2020. 142(17), 7968-7975).

The strategy that we control atomic diffusion by applying an electrical bias can be used to fabricate selectively two different heterostructures in the same material system. Thus, we believe that the results can provide a new perspective to synthesis individual nanostructures with a diffusion-controlled reaction and also throw light on the micro-mechanism of solid-solid processes.

We have added some discussion in the revised manuscript.

Supplementary Figure 9. Number of nanowires that were observed for each of the voltage conditions.

Supplementary Figure 11. Schematic illustration showing the difference of the mechanisms between our previous work (*Nature Communications*, 2017, 8, 14889) and current work. (a) The mechanism of the previous work “electrically driven cation exchange”. The Cu electrode was electro-ionized under the influence of a high voltage (larger than the standard electrode potential of Cu, 0.339V), after which Cu^+ migrated into the CdS nanorod along the direction of the electric field and reacted to form Cu_2S . (b) The mechanism of our current work “atomic diffusion driven by electron-wind force”. A low voltage was applied to the Ag-Te system, as a result, Ag or Te atoms could not be ionized electrochemically. Regardless the direction of electric field, the Ag or Te atoms always diffuse opposite to the direction of electric field and then spontaneously react into Ag_2Te . (c) A summary of differences between the two works.

REVIEWER COMMENTS

Reviewer #1 (Remarks to the Author):

In their revised manuscript, the Authors have presented additional data to show that electromigration is the primary force leading to the formation of Ag-Ag₂Te-Te heterostructures when Ag and Te nanowires are contacted under an applied bias. They show that minimal changes occur when the nanowires are irradiated by an electron beam in the absence of a bias and that they observe bias-dependent heterostructure formation on a large number of nanowires. I believe their revised manuscript adequately addresses the comments made by both Reviewers concerning the mechanism of heterostructure formation. I believe the revised manuscript is now suitable for publication in Nature Communications.

Reviewer 2 raised concerns about the novelty and significance of the work. The response of the Authors primarily focuses on how the current work is different from an earlier manuscript they have published in Nature Communications rather than describing the novelty and significance of the current work in its own right. I will defer to Reviewer 2 as to whether their original concerns on the novelty and significance have been adequately addressed.

Reviewer #2 (Remarks to the Author):

Thanks for the reply. I am convinced about their proposed mechanism for the electromigration-driven silver telluride heterostructures formation in both electric-field directions across the metal junction. The authors well-clarified why the observed phenomena described in the paper is different from the electrochemically-generated heterostructure formation (Supplementary Fig. 11 and related discussion). However, I still argue that the data discussed in the manuscript provide a limited view and the proposed model would be over-simplified.

In addition, I cannot see the significant novelty, which electron microscopy and electrochemistry can bring compared to other chemical process to synthesize nanostructured silver telluride. A quantitative approach for the atomic scale control of the thickness and interface of Ag₂Te/Te(Ag) structure over the electric field (not addressed other than the bias direction) and time (partially addressed) is also missing.

(i) As in the recent JACS paper, which the authors cited in response to the referee's comments (<https://doi.org/10.1021/jacs.0c02137>), the silver telluride nanowire can be easily formed under the TEM e-beam irradiation without any voltage applied when silver and tellurium nanowires are in close proximity. In this case, Ag⁺ ion migration is the primary reason for the silver telluride formation. This is in line with referee #1's comment and contrary to what the authors observed in the in-situ electrochemical TEM holder (Supplementary Fig. 8). It would be better for this paper to provide a more realistic picture of the whole conditions in which the electromigration-driven reaction happens vs the e-beam irradiation-driven reaction happens.

(ii) Supplementary Fig. 9 is just statistics that the authors did some experiments. I have not seen any other TEM images than those from the reactions at 0.3V and -0.8V bias, although they claim they conducted the experiments over the wide voltage. Do you observe the same phenomenon for all the number of nanowires observed across the electric field? Is there any quantitative voltage-dependent kinetics?

(iii) In the presented TEM images of the heterostructures, the diameter of the anode side is always smaller than their cathode side. Since the current density is important for the electromigration mechanism, what happens if their diameters are similar or the anode side is larger? Furthermore, in Fig. 1C, the Te nanowire on the cathode side shows also rough surfaces. Is there no heterostructure formed on the Te nanowire surface at all, not in the given experimental condition, but over the whole range of electric field applied?

RESPONSE TO REVIEWER COMMENTS

Reviewer #1 (Remarks to the Author):

In their revised manuscript, the Authors have presented additional data to show that electromigration is the primary force leading to the formation of Ag-Ag₂Te-Te heterostructures when Ag and Te nanowires are contacted under an applied bias. They show that minimal changes occur when the nanowires are irradiated by an electron beam in the absence of a bias and that they observe bias-dependent heterostructure formation on a large number of nanowires. I believe their revised manuscript adequately addresses the comments made by both Reviewers concerning the mechanism of heterostructure formation. I believe the revised manuscript is now suitable for publication in Nature Communications.

Reviewer 2 raised concerns about the novelty and significance of the work. The response of the Authors primarily focuses on how the current work is different from an earlier manuscript they have published in Nature Communications rather than describing the novelty and significance of the current work in its own right. I will defer to Reviewer 2 as to whether their original concerns on the novelty and significance have been adequately addressed.

Response: Thank you for your positive comments that “the revised manuscript is now suitable for publication in *Nature Communications*”. About the novelty and significance, we presented a strategy where we tailor atomic diffusion by applying an electrical bias inside a TEM that enables us to directly observe the formation of different heterostructures with atomic resolution. A mechanism of electrically driven atomic diffusion is proposed, and based on the mechanism, controlled selective synthesis of individual heterostructure could be achieved by simply adjusting the current and local temperature, which is critical to tailor nanostructures and for the construction of future nanodevices. We have added some descriptions in the revised manuscript.

Reviewer #2 (Remarks to the Author):

Thanks for the reply. I am convinced about their proposed mechanism for the electromigration-driven silver telluride heterostructures formation in both electric-field directions across the metal junction. The authors well-clarified why the observed phenomena described in the paper is different from the electrochemically-generated heterostructure formation (Supplementary Fig. 11 and related discussion). However, I still argue that the data discussed in the manuscript provide a limited view and the proposed model would be over-simplified.

In addition, I cannot see the significant novelty, which electron microscopy and electrochemistry can bring compared to other chemical process to synthesize

nanostructured silver telluride. A quantitative approach for the atomic scale control of the thickness and interface of Ag₂Te/Te(Ag) structure over the electric field (not addressed other than the bias direction) and time (partially addressed) is also missing.

(i) As in the recent JACS paper, which the authors cited in response to the referee's comments (<https://doi.org/10.1021/jacs.0c02137>), the silver telluride nanowire can be easily formed under the TEM e-beam irradiation without any voltage applied when silver and tellurium nanowires are in close proximity. In this case, Ag⁺ ion migration is the primary reason for the silver telluride formation. This is in line with referee #1's comment and contrary to what the authors observed in the in-situ electrochemical TEM holder (Supplementary Fig. 8). It would be better for this paper to provide a more realistic picture of the whole conditions in which the electromigration-driven reaction happens vs the e-beam irradiation-driven reaction happens.

Response: Thank you for your comments. The referenced article (*J. Am. Chem. Soc.* 2020, 142, 7968–7975) reported that the silver telluride (Ag₂Te) nanowire is formed under the electron beam irradiation when silver (Ag) and tellurium (Te) nanowires are in close proximity. Both Ag and Te nanowires are placed parallelly on a film, and the Ag species sputtered by electron beam would migrate in all directions on the film surface to react with Te, resulting in formation of Ag₂Te nanowires. However, in our experiments, Ag and Te nanowires are suspended and point contacted, electron beam-excited Ag species can only migrate into Te nanowire through the contact point, which will seriously limit the atomic diffusion and the formation of Ag₂Te. That's why we did not observe obvious Ag₂Te formed under electron beam irradiation without any voltage applied when Ag and Te nanowires were point contacted (Supplementary Figure 8). Besides, the diffusion of excited species is strongly dependent on the direction of electric field. These results confirm that electrically driven diffusion is the primary reason for the formation of Ag₂Te in our work, and the influence of electron beam can be ignored. We have added some discussions in the revised manuscript.

(ii) Supplementary Fig. 9 is just statistics that the authors did some experiments. I have not seen any other TEM images than those from the reactions at 0.3V and -0.8V bias, although they claim they conducted the experiments over the wide voltage. Do you observe the same phenomenon for all the number of nanowires observed across the electric field? Is there any quantitative voltage-dependent kinetics?

Response: Thank you for your comment. Some other TEM image series have been added (Supplementary Figure 10 and 11), showing the reaction between Ag and Te nanowires at various bias (-0.2 V, -0.3V, -0.5V, -1.0 V, 0.2 V, 0.3 V, 0.5 V, 0.7 V). Similar phenomena were observed in these experiments in which the nanowires were under different bias.

We analyzed the formation of Ag₂Te and did not find obvious dependency between the applied voltage and the growth rate of Ag₂Te (Supplementary Figure 14 in the

revised manuscript). It may be attributed to the contact conditions which will affect the atomic diffusion and subsequent reaction. It's hard to ensure same contact in separate experiment and the condition is difficult to be quantified inside the TEM, which makes challenges to reveal an quantitative voltage-dependent kinetics from various Ag-Te systems. Some discussions have been added in the revised manuscript.

Supplementary Figure 10. Electrically driven diffusion of Te atoms under the bias of Ag to Te.

Supplementary Figure 11. Electrically driven diffusion of Ag atoms under different conditions.

Supplementary Figure 14. The velocity of the growth of Ag_2Te layers (pentagram, when Te diffuse into Ag by surface diffusion) and the velocity of the transformation (hexagon, when Ag diffuse into Te by bulk diffusion) under different bias.

(iii) In the presented TEM images of the heterostructures, the diameter of the anode side is always smaller than their cathode side. Since the current density is important for the electromigration mechanism, what happens if their diameters are similar or the anode side is larger? Furthermore, in Fig. 1C, the Te nanowire on the cathode side shows also rough surfaces. Is there no heterostructure formed on the Te nanowire surface at all, not in the given experimental condition, but over the whole range of electric field applied?

Response: Thank you for your comment. We have added Supplementary Figure 10 and 11 in the revised manuscript. As shown in Supplementary Figure 10 and 11, no matter which side is larger, similar phenomena were observed. The core-shell structures were obtained in the condition that Ag nanowire was on the anode side, while Ag_2Te -Te heterostructures formed when the direction of bias was changed. However, the diameter of the nanowires may affect the formation rate of Ag_2Te . As shown in Supplementary Figure 10, the thinner Ag nanowire (Supplementary Figure 10(c), diameter: 68 nm) has a faster reaction velocity under a same bias of 0.5 V compared to the thicker nanowire (Supplementary Figure 10(d), diameter: 79nm). Figure 2 and Supplementary Figure 10(b) (diameters: 78 nm and 42 nm, respectively) also show that the reaction of thinner nanowires is faster than that of the thick nanowires. Some discussions have been added in the revised manuscript.

We have reanalyzed the Supplementary Video 4 of the Figure 1c and found that the surface of Te is smooth before reaction while rough after reaction, which confirmed the formation of heterostructure under a bias of Te to Ag (Figure 4). Some other experiments in Supplementary Figure 11 can also confirm the formation of heterostructures.

Figure 4. (a-f) The formation of a Ag_2Te -Te binary heterostructure nanowire.

REVIEWERS' COMMENTS

Reviewer #2 (Remarks to the Author):

With the revision, now the authors provide sufficient supporting data to make this paper solid. For example, I do like the supplementary Fig.14.

On the other hand, I wish the paper would have revised to claim clear novelty in the context of already well-established nanomaterials synthesis. Nowadays, even solution-based cationic exchange reaction produces very well-defined heterostructured nanocrystal (<https://pubs.acs.org/doi/10.1021/acs.chemmater.0c01388>). Here I cannot see the merit of electrochemistry to bring WOW to nanomaterials synthesis.

For this reason, I am not totally convinced if this work is an original and unique contribution to nanomaterials synthesis enough to be published in Nature Communications. However, I evaluate this paper is solid enough to be published elsewhere.

RESPONSE TO REVIEWER COMMENTS

Reviewer #2 (Remarks to the Author):

With the revision, now the authors provide sufficient supporting data to make this paper solid. For example, I do like the supplementary Fig.14.

On the other hand, I wish the paper would have revised to claim clear novelty in the context of already well-established nanomaterials synthesis. Nowadays, even solution-based cationic exchange reaction produces very well-defined heterostructured nanocrystal (<https://pubs.acs.org/doi/10.1021/acs.chemmater.0c0-1388>). Here I cannot see the merit of electrochemistry to bring WOW to nanomaterials synthesis.

For this reason, I am not totally convinced if this work is an original and unique contribution to nanomaterials synthesis enough to be published in Nature Communications. However, I evaluate this paper is solid enough to be published elsewhere.

Response: Thank you for your comments and recognition of our work. About the novelty and significance, the electrical methods of tailoring atomic diffusion can achieve selective customization for individual nanostructures and the location of hetero-interface can also be controlled, compared to the solution-based reaction. According to your comment, we have made a revision about the descriptions of the novelty in the revised manuscript.